# Obesity in Adolescents Who Skip Breakfast Is Not Associated with Physical Activity

**DOI:** 10.3390/nu11102511

**Published:** 2019-10-18

**Authors:** Sara Sila, Ana Ilić, Marjeta Mišigoj-Duraković, Maroje Sorić, Ivan Radman, Zvonimir Šatalić

**Affiliations:** 1Referral Centre for Paediatric Gastroenterology and Nutrition, Children’s Hospital Zagreb, Klaićeva 16, 10000 Zagreb, Croatia; 2Department of Food Quality Control, Faculty of Food Technology and Biotechnology, University of Zagreb, Pierottijeva ul. 6, 10000 Zagreb, Croatiazsatalic@pbf.hr (Z.Š.); 3Department of Kinesiological Anthropology and Methodology, Faculty of Kinesiology, University of Zagreb, Horvaćanski zavoj 15, 10000 Zagreb, Croatia; marjeta.misigoj-durakovic@kif.hr (M.M.-D.); maroje.soric@kif.hr (M.S.); 4Faculty of Sport, University of Ljubljana, Gortanova ulica 22, 1000 Ljubljana, Slovenia; 5Department of Kinesiology of Sports, Department of Kinesiological Anthropology and Methodology, Faculty of Kinesiology, University of Zagreb, Horvaćanski zavoj 15, 10000 Zagreb, Croatia; ivan.radman@kif.hr

**Keywords:** eating habits, obesity, adolescents, physical activity

## Abstract

It has been hypothesized that breakfast consumption is generally associated with healthy lifestyle behaviors, such as increased physical activity. The aim of this study was to examine the relations between breakfast consumption, adiposity measures and physical activity among adolescents. This investigation is a part of the Croatian physical activity in adolescence longitudinal study (CRO-PALS). This investigation is based on 802 participants (48% girls and 52% boys), aged between 15.2 and 16.0 years. Physical activity level and sedentary behaviors were assessed using the SHAPES questionnaire. Adiposity measures included the sum of four skinfolds, and multi-pass 24-h recall was used as the dietary assessment method. Participants who consumed breakfast had significantly lower body fat % (*p* = 0.011 for boys; *p* ≤ 0.001 for girls) compared to breakfast non-consumers. Physical activity has no mediating effect in the association of breakfast consumption on adiposity in boys (Sobel’s t = −0.541; *p* = 0.588) and girls (Sobel’s t = 1.020; *p* = 0.307). Breakfast consumption was negatively associated with adiposity only in the boys at the highest tertile of physical activity (*p* = 0.04). Physical activity has no mediating effect on the associations between breakfast consumption and adiposity, but has a moderation effect only in the most active boys. Breakfast consumption might exert beneficial effects only in the most active male adolescents, but not in the inactive ones.

## 1. Introduction

Obesity in children and adolescents is one of the greatest public health concerns of the twenty-first century [1]. It was found that between 25% and 58% of adolescent who are overweight will become overweight adults, and between 24% and 90% of obese adolescents will become obese adults [2], which shows that primary prevention is of utmost importance. 

Daily eating patterns such as eating frequency and distribution of eating occasions throughout the day have received considerable attention for their potential role in obesity prevention [3], with breakfast consumption remaining most commonly reported as the diet factor involved in obesity development [4]. Recent studies have shown that skipping breakfast is associated with weight gain and increased risk of obesity, but a direct cause-effect relationship has yet to be established [4]. 

It has been emphasized recently that if habitual breakfast eaters do, indeed, have a lower body mass index (BMI), this implies that regular breakfast consumption is related to reduced energy intake and/or increased energy expenditure over time. It has been noticed previously that typically, adolescents who consume breakfast have a lower BMI despite having higher energy intake [5]. These findings suggest that breakfast consumption is probably associated with higher energy expenditure, rather than restricted consumption [6]. Indeed, several cross-sectional studies have reported positive associations between breakfast consumption and physical activity and physical fitness, but only a small number of the studies are controlled for energy expenditure [4]. 

Understanding how breakfast consumption and other factors associated with breakfast consumption could affect energy balance will help understand the relationship between breakfast consumption and adiposity, as well as help to develop appropriate measures for obesity prevention. Therefore, the purpose of our study was to examine the relations between breakfast consumption, adiposity measures and physical activity in a sample of adolescents. Using a mediation framework, we have also explored the contributing role that physical activity (PA) time plays in association with breakfast consumption and adiposity. Moderation effect of PA on the association between breakfast consumption and adiposity was estimated by tertiles of physical activity. We hypothesized that adolescents who skip breakfast would have higher measures of adiposity and that this association would be partially explained by lower levels of physical activity.

## 2. Materials and Methods 

### 2.1. Participants and Settings

This investigation is a cross-sectional study that is a part of the CRO-PALS study (Croatian physical activity in adolescence longitudinal study), a longitudinal observational study on a randomly selected sample of adolescents set in the city of Zagreb (Croatia). Specifically, stratified two-stage random cluster sampling was used to select the participants for this study using sets of random numbers generated in Excel. First, all 86 secondary schools in Zagreb area were stratified by type: grammar schools/vocational schools/private schools. Next, at the first stage of random selection of clusters, based on the proportion of different types of schools and the average number of students in the first grade per school, 13 public (8 vocational and 5 grammar schools) and 1 private school (grammar school), were selected with total of 2827 students enrolled. During the second stage of randomization, half of the first grade classes in each of the selected schools were randomly selected. Finally, all 1408 students enrolled in the selected classes were approached, regardless of their health condition and 903 agreed to participate (response rate = 64%).

This investigation is based on 802 participants (48% girls and 52% boys) aged between 15.2 and 16.0 years with complete data on nutrition and physical activity. All measurements were performed during spring 2014. Because boys and girls differ markedly both in physically activity level and adiposity, all analyses were stratified by gender.

After being informed about the aim and procedures of the study both the participants and their parents gave their informed written consent. The study was performed according to the Declaration of Helsinki and all the procedures were approved by the Ethics Committee of the Faculty of Kinesiology, University of Zagreb (No. 1009-2014).

### 2.2. Anthropometry

Participants were weighed barefoot in their shorts and T-shirts with a pre-calibrated portable digital scale to the nearest 0.1 kg. Body height was taken to the nearest 0.1 cm using an anthropometer. BMI was then calculated as body weight in kilograms divided by body height in meters squared (kg/m^2^). Extended age- and sex-specific BMI cut-off points proposed by the International Obesity Task Force were used to distinguish between normal-weight and overweight or obese children [7].

Skinfold thickness were taken to the nearest 0.1 mm using a Harpenden skinfold calliper (British Indicators, West Sussex, UK) on the right side of the body [8]. Skinfolds were measured at four sites as follows: (1) triceps—halfway between the acromion process and the olecranon process, (2) subscapular—about 20 mm below the tip of the scapula at an angle of 45° to the lateral side of the body, (3) suprailiac—above the iliac crest at the level of the anterior axillary line, and (4) calf—at the level of maximum calf circumference, on the medial aspect of the calf. All skinfold measures were taken in triplicate and median values were retained for analysis. The sum of 4 skinfolds (S4SF) was chosen as an indicator of adiposity as skinfolds were repeatedly shown to better reflect body fatness compared to BMI [9]. Skinfold measurements on all participants were performed by a single, skilled lab technician.

### 2.3. Physical Activity (PA)

Physical activity level and sedentary behaviors were assessed using the School Health Action, Planning and Evaluation System (SHAPES) questionnaire [10]. The physical activity module of the questionnaire includes two items requesting 7-day recall of moderate intensity physical activity (MPA) and vigorous intensity physical activity (VPA). Participants had to indicate the number of hours (0–4 h) and 15-min increments (0–45 min) that MPA and VPA were performed for each day of the previous week. Average daily physical activity energy expenditure (PAEE) over the investigated 7-day period was calculated as proposed by Wong et al. [10], assuming an average intensity of 4 metabolic equivalent of task (MET) for MPA and 7 METs for VPA. Sedentary behaviors are assessed through 7 items examining the average time spent: (1) playing computer/video games, (2) television viewing, (3) browsing the Internet, (4) homework and studying, (5) listening to music, (6) reading and (7) playing instruments. As for physical activity, responses are provided by indicating the number of hours and minutes in 15-min increments [10].

The SHAPES questionnaire has been previously shown to be a valid and reliable instrument for assessing PA and sedentary behavior (SB) in primary and secondary school children [10].

### 2.4. Dietary Assessment

Multi-pass 24-h recall (5 steps) was the dietary assessment method performed by trained interviewers, and the referent day was a typical workday. The interviews were not performed on Mondays, and the reference to Friday was avoided as the day that resembles weekend dietary habits. Portion size was estimated using national food portion booklet providing three (small, medium and large) photos for each item [11]. Subjects were contacted afterwards if clarification or more details were necessary. Participants who had significantly reduced their food intake on the day prior to the dietary assessment due to health-related or any other reasons were excluded from the analysis. Conversion of collected data to energy and nutrient intakes was performed using national food composition tables, complemented with a Danish database for food not found in the Croatian tables [12,13]. Breakfast was defined as the intake of foods or beverages before 10:00, containing a minimum of 250 kcal. Among several different definitions of breakfast [14], we opted for a definition related to energy intake driven by suggestions for defining eating occasions using minimal energy intake [15]. In Croatian dietary guidelines for children [16], a minimum of 10% of total energy intake is considered a meal instead of a snack. Since average energy requirements in adolescents are estimated at 2500 kcal/day, we set 250 kcal as the minimal energy intake to be taken on a single eating occasion. Participants were defined as breakfast consumer or breakfast non-consumer based on the described criteria. 

### 2.5. Statistical Analyses

Data on participants’ descriptive characteristics, breakfast consumption, adiposity and PA level were processed using SPSS software, version 24.0 (IBM, New York, NY, USA), and presented as means ± SD and medians (interquartile range) for normally distributed and not-normally distributed continuous variables, respectively, and as percentages for categorical variables. Initially, differences among girls and boys were assessed using T-test and Median test, with dependence on the variable’s nature. For the following analyses, participants were first stratified by sex and then compared in adiposity measures and daily behavior patterns based on their breakfast consumption habit. First, within an unadjusted model, differences between breakfast consumers and non-consumers were examined via analysis of variance (ANOVA) and Kruskal-Wallis ANOVA, as appropriate. Afterwards, the differences between breakfast consumption groups were tested through analysis of covariance (ANCOVA), and adjusted for age and socio-economic status (SES) as covariates. In addition, T-test and Median test were used to assess possible differences among breakfast consumption groups for age and SES. 

To explore the association between breakfast consumption habit and subcutaneous fat, as well as the possible mediation effect of PA, linear regression analysis was applied first in a simple model adjusted for age, SES and sleep time, and subsequently in two models further adjusted for: (1) weekly volume of PA, and (2) weekly volume of PA + sedentary time. For ease of interpretation, unstandardized regression coefficients B (and 95% confidence intervals) are presented. The significance of the mediation effect of PA on the association between breakfast consumption and subcutaneous fat was formally assessed via Sobel test [17]. Finally, in order to investigate the possible moderation effect of PA on the association between breakfast consumption and adiposity, participants were first divided by sex-specific tertiles of moderate-to-vigorous physical activity (MVPA), and then all the linear regression models described previously were repeated in each tertile of PA separately. For all conducted analyses, the level of significance was set at *p* < 0.05.

The study was performed according to the Declaration of Helsinki and all the procedures were approved by the Ethics Committee of the Faculty of Kinesiology, University of Zagreb (No. 1009-2014).

## 3. Results

The descriptive statistics are summarized in Table 1. In total, 39.5% of participants did not consume breakfast. While boys had significantly higher energy expenditure (10 kcal/kg/day vs. 7 kcal/kg/day, *p* < 0.001) and were more likely (66.5% vs. 53.9%, *p* < 0.001) to consume breakfast, they had higher prevalence of overweight (16.5% vs. 14.1%, *p* <0.001) and obesity (6.0% vs. 2.1%, *p* < 0.001) compared to girls.

Differences in BMI, adiposity measures and PA between breakfast consumers and non-consumers are presented in Table 2. Participants who consumed breakfast had significantly lower BMI (21.5 ± 3.2 vs. 22.4 ± 4.2, *p* = 0.022 for boys; 20.8 ± 3.1 vs. 22.2 ± 3.1, *p* ≤0.001 for girls) and body fat % (16.1 ± 7.7 vs. 18.2 ± 8.9, *p* = 0.011 for boys; 23.0 ± 4.1 vs. 24.7 ± 4.1, *p* ≤0.001 for girls) but higher total daily energy intake (2544 vs. 1915 kcal/d, *p* ≤0.001 for boys; 1726 vs. 1372, *p* ≤0.001 for girls) compared to breakfast non-consumers. 

The difference in overweight prevalence between boys who skipped or consumed breakfast just failed to reach significance (18.6% for non-consumers vs. 15.5 for consumers, *p* = 0.058), while lower overweight prevalence has been observed in girls who consumed breakfast compared to those who did not (9.7% vs. 19.3%, *p* < 0.001). 

MVPA, sedentary time and sleep time did not differ between boys who were breakfast consumers or non-consumers, while girls who skipped breakfast had significantly less hours of sleep compared to the girls who consume breakfast (7.21 ± 1.56 vs. 7.55 ± 1.65, *p* = 0.041). When the model was adjusted for SES and age, the sum of the skinfolds in boys who consumed compared to those who did not consume breakfast was significantly lower (30 vs. 33 mm, *p* = 0.020). No significant difference in duration of sleep between the girls who skipped or consumed breakfast was found when the model was adjusted for SES and age. Results of other measurements remained the same in adjusted model.

Table 3 represents simple and adjusted associations between breakfast consumption and adiposity in both genders. Results show that there was significant negative association between breakfast consumption and adiposity in both boys (B = −.917, 95% Cl −9.041–−0.793, p = 0.02) and girls (B = −4.138, 95% Cl −7.499–−0.777, p = 0.016) in the sample adjusted for age, SES and average sleeping time. These associations remained unchanged after adjustment for either MVPA, or MVPA and ST.

Mediation analysis found no mediating effect of PA on the association of breakfast consumption on adiposity in both boys (t = −0.541; *p* = 0.588) and girls (t = 1.020; *p* = 0.307) (Figure 1).

For the analysis of the possible moderation effect of PA on the association between breakfast consumption and adiposity, the participants were divided into tertiles of MVPA (Figure 2). In boys, there was no association between adiposity and breakfast consumption at the lowest tertile of MVPA. In the second tertile of MVPA, decreased adiposity measures were found, although association with breakfast consumption was not significant. Only in the most active boys was there significant association between breakfast consumption and lower adiposity (*p* = 0.04). An identical pattern of moderation effect was seen also in girls, although a slightly wider CI was noted and no significant effect was noticed.

## 4. Discussion

This study found no association between breakfast consumption and physical activity in our sample, which was confirmed by a formal mediation model. Although, when grouped by tertiles of physical activity, in boys at the highest tertile of PA, breakfast consumption was negatively associated with adiposity.

Our finding that breakfast consumers have higher total daily energy intake but lower adiposity measures is in accordance with previous research [4,5]. One of the hypotheses is that breakfast consumption is important for the regulation of energy intake, and therefore, skipping breakfast could result in higher energy intake at other meals compared with when breakfast is consumed [18]. A review by Blondin et al. [4] has concluded that there is a possible protective role of breakfast consumption in preventing excess adiposity during childhood and adolescence, although a causal relationship is not a certainty. It has been repeatedly emphasized that residual confounding could bias results of observational studies, since breakfast consumption is generally associated with healthy lifestyle behaviors. Therefore, the attention has been switched to possible confounding and mediating variables, such as eating frequency, breakfast content and energy expenditure. In a randomized cross-over study, Zakrzewski-Fruer et al. [19] examined the acute effect of breakfast consumption and omission on free-living energy intake and PA in adolescent girls. Breakfast manipulation did not affect time spent sedentary or in PA, and the study group concluded that there was no evidence that breakfast consumption induces compensatory changes in PA. 

This study found a negative association between breakfast consumption and adiposity measures after the adjustment for MVPA or MVPA and ST. No association between breakfast consumption and MVPA was found. The hypothesis of this study has been based on a recent review by Blondin et al. [4], which indicated that physical activity could be a potential mediator of the relationship between breakfast consumption and adiposity. Results of previous studies are inconclusive. In a study by Schembre et al. [6], where objective measures of physical activity have been used, more frequent breakfast eaters spent on average 30% more minutes a day in MVPA than the those in the low frequency breakfast group. Contrary to this study, in a sample of 2148 adolescents from ten European cities, the HELENA study found no association of breakfast consumption with either objectively measured or self-reported physical activity [20]. The same has been observed in a study by Corder et al. [21]. The reason for this discrepancy could be explained by different methodologies used to assess physical activity. Results of our study are in concordance with the results of the HELENA study, where similar methodologies and sample to ours were used. 

Using a formal mediation test, we wanted to identify and explain the mechanism that underlies an observed relationship between breakfast consumption and adiposity via the inclusion of a MVPA as a mediator variable. We assumed that eating breakfast alone does not affect the occurrence of obesity, but that in this relationship, the mediation role is physical activity. We speculate that those who consume breakfast are not less obese because they consume breakfast, but because they are more physically active, which requires higher caloric intake. Higher caloric intake could potentially be satisfied by breakfast consumption. However, the mediation test model, which was age-, SES- and sleep time-adjusted, confirmed that MVPA is not a mediating factor in this association neither in boys nor girls. Schembre et al. [6] were unable to detect significance in the mediation test model because of the limited sample size, but they suggest that objectively assessed MVPA may partially mediate the relationship between breakfast consumption and adiposity in predominantly overweight Latin and African American girls. In contrast, Albertson et al. [22], in their longitudinal study, showed that PA mediates the association between breakfast consumption and BMI over time. The inconsistency in results can be result of different study designs, sample sizes, methods for estimation of breakfast consumption and PA, criteria of adiposity, baseline anthropometric characteristics, race and socioeconomic status of the sample. 

It can be noticed that from the lowest to the highest tertile of PA, the negative association between breakfast consumption and adiposity increases in both boys and girls. However, statistically significant negative associations between breakfast consumption and adiposity have been observed only in the most active boys. Generally, this means that physical activity moderates the relationship between breakfast consumption and obesity. Discordant to this, Karatzi et al. [23] have found that children at the lowest quartile of PA had a positive association between the calories consumed at dinner, which was associated with breakfast skipping and BMI. On the contrary, this association was inverse for children being at the highest quartile of physical activity levels. Also, as mentioned above, the results from the same study show that male breakfast consumers, compared to breakfast non-consumers, had a greater total daily energy intake, which suggests that increased PA, especially vigorous PA, seems to compensate for the negative effects of excessive caloric intake throughout the day. Therefore, the mediation of regular vigorous PA on the association between breakfast consumption and adiposity will require further consideration.

Strengths of this study include a reasonably large sample size and inclusion of mediating variables such as PA, sleeping time and SES. Post-hoc analysis performed using the statistical power analysis program G*Power for Windows 3 [24], as demonstrated this study, was able to detect a medium effect size (f^2^ = 0.15 [25] in both girls and boys with the power (1–β error probability) of >0.999 when setting the significance level at *p* = 0.05. Next, we have used subcutaneous fat as a measure of adiposity for the analysis of association between breakfast consumption and adiposity, since BMI is an inaccurate measure of adiposity in active children due to their larger muscle mass [9]. 

This study has also several limitations worth addressing. This is an observational cross-sectional study, so we could not establish causal inferences. Another limitation of the study is that PA assessment was based on self-reported data. Low validity of self-report possibly led to an underestimation of mediating effect of PA. On the other hand, as self-reported PA is reasonably accurate in ranking individuals, the moderation effect should not be affected. Finally, breakfast consumption was assessed by using 24-h dietary recall, which might not be representative of habitual diet at an individual level, but it should be adequate for surveying intake in a large sample.

## 5. Conclusions

In this study, no association between breakfast consumption and physical activity was noted, which was further confirmed by the formal mediation model. When grouped by tertiles of physical activity, the associations of breakfast consumption and lower adiposity was not present in low active groups of male adolescents. This could imply that breakfast consumption could exert beneficial effects only in at the most active male adolescents, but not in the inactive ones. Future relevant studies that will use objective methods of physical activity assessment in large samples as well as additional confounders, such as number of eating occasions and total daily energy intake, are needed.

## Figures and Tables

**Figure 1 nutrients-11-02511-f001:**
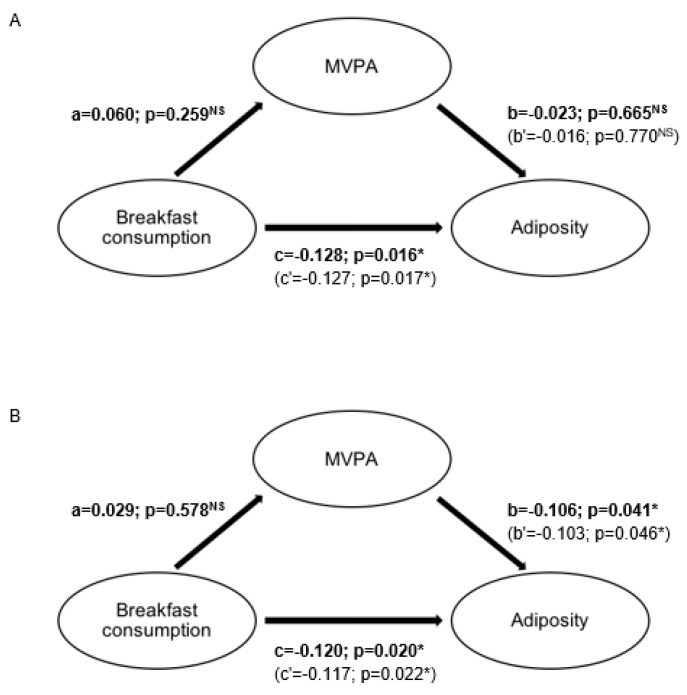
Mediation model of the effect of moderate-to-vigorous physical activity on the association between breakfast consumption and adiposity in girls (**A**) and boys (**B**). MVPA—moderate-to-vigorous physical activity; NS—non-significant; * significant at *p* <0.05. Values in boldface type represent age-, socioeconomic status- and sleep time-adjusted associations between breakfast consumption and MVPA (total effect, a), MVPA and adiposity (total effect, b) and breakfast consumption and adiposity (total effect, c). Values in parentheses represent age-, socioeconomic status- and sleep time-adjusted associations between breakfast consumption and adiposity with control for the hypothesized mediator, MVPA (direct effect, c’), and between MVPA and adiposity with control for breakfast consumption (direct effect, b’). The indirect or mediation effect (ab’) was −0.001 and −0.003 in girls and boys, respectively, and was deemed nonsignificant as assessed with the Sobel test (t = 1.020, *p* = 0.31 in girls and t = −0.54, *p* = 0.59 in boys).

**Figure 2 nutrients-11-02511-f002:**
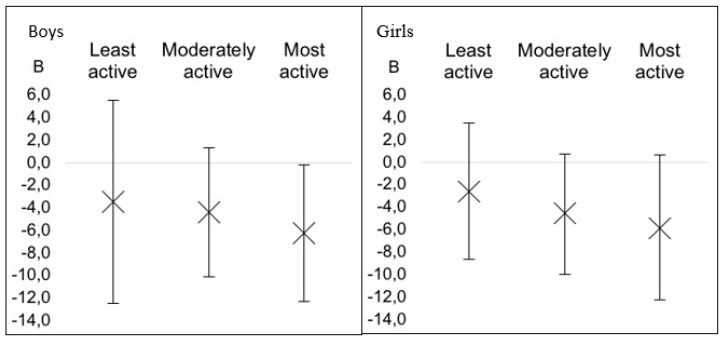
Associations between breakfast consumption and adiposity in boys and girls stratified by tertiles of physical activity. Unstandardized regression coefficients B and 95% confidence intervals are shown. Association significant at *p* < 0,05. Average moderate-to-vigorous physical activity across tertiles—Boys: least active 4.6 ± 1.7 (*p* = 0.45), moderately active 9.8 ± 1.4 (*p* = 0.13) and most active 19.2 ± 5.5 (*p* = 0.04) kcal/kg/day. Girls: least active 3.4 ± 1.3 (*p* = 0.40), moderately active 7.3 ± 1.2 (*p* = 0.09) and most active 15.7 ± 5.1 (*p* = 0.07) kcal/kg/day.

**Table 1 nutrients-11-02511-t001:** Basic descriptive characteristics of participants ^1^.

Adolescent’s Characteristics	Total (*n* = 802)	Boys (*n* = 418)	Girls (*n* = 384)	*p* *
**Age (yr.)**	15.6 ± 0.4	15.7 ± 0.4	15.6 ± 0.4	0.002
**Energy intake**				
Breakfast (kcal)	311 (124–505)	371 (140–593)	269 (102–437)	<0.001
Total daily intake (kcal)	1907 (1409–2660)	2339 (1759–3079)	1534 (1195–2089)	<0.001
**Eating occasions**	4 (4–5)	4 (4–5)	4 (3–5)	0.009 *
**Breakfast consumption**				
Non-consumers	39.5%	33.5%	46.1%	<0.001
Consumers	60.5%	66.5%	53.9%	
**Overweight prevalence**	15.2%	16.5%	14.1%	<0.001
**Obesity prevalence**	4.1%	6.0%	2.1%	<0.001
**Body fat (%)**	20.1 ± 7.5	16.8 ± 8.2	23.8 ± 4.2	<0.001
**Sum of skinfolds (mm)**	39 (29–54)	31 (25–41)	48 (38–60)	<0.001
**Fat free mass (kg)**	50.9 ± 8.4	56.5 ± 6.6	44.9 ± 5.5	<0.001
**BMI (kg/m^2^)**	21.6 ± 3.4	21.8 ± 3.6	21.4 ± 3.1	0.103
**MVPA (kcal/kg/day)**	9 (5–14)	10 (6–15)	7 (5–12)	<0.001
**Sedentary time (min/day)**	413.8 ± 197.7	405.0 ± 196.2	420.5 ± 194.8	0.266
**Screen time (min/day)**	223 (146–339)	240 (163–351)	214 (126–330)	<0.001
**Sleep time (h/day)**	7.5 ± 1.6	7.6 ± 1.5	7.4 ± 1.6	0.022
**Socioeconomic status**				
Much lower than average	13.5%	15.5%	12.0%	0.330
Lower than average	34.1%	34.2%	33.9%	
Average	45.9%	45.0%	46.2%	
Higher than average	5.8%	5.0%	6.7%	
Much higher than average	0.7%	0.3%	1.1%	
**School type**				
Grammar	38.2%	30.9%	47.4%	<0.001
Vocational	61.8%	69.1%	52.6%	

BMI—body mass index; MVPA—moderate-to-vigorous physical activity. ^1^ Mean values (± standard deviations) are for normally distributed, median values (and interquartile range) are for not-normally distributed continuous variables and percentages (n) are for categorical variables. * Group differences were tested using T-test, Chi-square test and Median test.

**Table 2 nutrients-11-02511-t002:** Differences in adiposity measures and physical activity (PA) between breakfast consumption groups ^1^.

Adolescent’s Characteristics	Non-Consumers	Consumers	Unadjusted Model ^2^ *p*	Adjusted Model ^3^ *p*
**Boys (n = 418)**				
Body mass index (kg/m^2^)	22.4 ± 4.2	21.5 ± 3.2	0.022	0.018
Body fat (%)	18.2 ± 8.9	16.1 ± 7.7	0.011	0.006
Sum of skinfolds (mm)	33 (27–46)	30 (25–40)	0.072	0.020
Total daily energy intake (kcal)	1915 (1440–2656)	2544 (1962–3254)	<0.001	<0.001
Overweight prevalence (%)	18.6	15.5	0.080	0.058
MVPA (kcal/kg/day)	9 (6–14)	10 (6–15)	0.263	0.577
Sedentary time (min/day)	423 ± 198	396 ± 195	0.195	0.165
Screen time (min/day)	296 ± 167	274 ± 163	0.209	0.180
Sleep time (h/day)	7.64 ± 1.49	7.65 ± 1.48	0.936	0.861
**Girls (n = 384)**				
Body mass index (kg/m^2^)	22.2 ± 3.1	20.8 ± 3.1	<0.001	<0.001
Body fat (%)	24.7 ± 4.1	23.0 ± 4.1	<0.001	<0.001
Sum of skinfolds (mm)	52 (41–63)	44 (36–56)	<0.001	0.011
Total daily energy intake (kcal)	1372 (937–1848)	1726 (1384–2237)	<0.001	<0.001
Overweight prevalence (%)	19.3	9.7	<0.001	<0.001
MVPA (kcal/kg/day)	7 (4–12)	8 (5–13)	0.240	0.280
Sedentary time (min/day)	421 ± 187	420 ± 202	0.929	0.975
Screen time (min/day)	240 ± 133	241 ± 160	0.923	0.853
Sleep time (h/day)	7.21 ± 1.56	7.55 ± 1.65	0.041	0.054

MVPA—moderate-to-vigorous physical activity. ^1^ Mean values (± standard deviations) are for normally distributed, median values (and interquartile range) are for not-normally distributed continuous variables and percentages (n) are for categorical variables. Consumers were defined as those who had intake of foods or beverages before 10:00, containing a minimum of 250 kcal. ^2^ Group differences were tested using analysis of variance (ANOVA) for parametrical variables and Kruskal-Wallis ANOVA for non-parametrical variables (**p* < 0.05). ^3^ Adjusted for age and socioeconomic status. Group differences were tested using analysis of covariance (ANCOVA) (**p* < 0.05).

**Table 3 nutrients-11-02511-t003:** Simple and adjusted associations between breakfast consumption and adiposity in boys and girls.

Gender	Simple ^2^			Adjusted for MVPA ^3^			Adjusted for MVPA and ST ^4^		
	B	95% CI	*p*	B	95% CI	*p*	B	95% CI	*p*
**Boys (*n* = 418)**	−4.9	−9.0–−0.8	0.020	−4.8	−8.9–−0.7	0.022	−4.6	−8.7–−0.4	0.031
**Girls (*n* = 384)**	−4.1	−7.5–−0.8	0.016	−4.1	−7.5–−0.7	0.017	−4.1	−7.5–−0.7	0.018

Unstandardized regression coefficients B and 95% confidence intervals are shown. Consumers were defined as those who reported intake of foods or beverages before 10:00, containing a minimum of 250 kcal. ^2^ Simple is adjusted for age, socioeconomic status and average sleep time. ^3^ Adjusted for age, socioeconomic status, average sleep time and moderate-to-vigorous physical activity (MVPA) (continuous variable). ^4^ Adjusted for age, socioeconomic status, average sleep time, moderate-to-vigorous physical activity (MVPA) as a continuous variable and ST (sedentary time) as continuous variable.

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
