# Peer review of "Obesity in Adolescents Who Skip Breakfast Is Not Associated with Physical Activity"

_nutrients, 2019, doi:10.3390/nu11102511_

Round 1
Reviewer 1 Report
Though this is an interesting study, the limitation of 250kcal on what constitutes eating breakfast is potentially a methodological error. The methodology requires some expansion, though it is good to see that SES has been considered. The manuscript reaches a well considered conclusion and i am sure this paper will receive sufficient interest.
Specific comments:
Abstract: CRO-PALS needs to be defined. Please specify that 't' is related to Sobel's 't' to provide clarity.
Please clarify how the cohort was recruited, the children approached to participate were determined by a cluster randomisation process? Please discuss if there were any inclusion or exclusion criteria associated with selection of the cohort, was there any consideration of dieting habits, disordered eating behaviours, health conditions, medications etc.
Were any special considerations made to taking the anthropometric measurements of adolescents?
Were the skin fold measurements taken by a person with International Association of Kinanthropometry qualifications?
Please add in evidence to the literature as to why the sum of four skin folds is considered an appropriate measure of adiposity.
Were parents consulted regarding the child's food recall to ensure accuracy?
Why was an energy qualifier put on what constitutes breakfast? A healthy and filling breakfast can quite easily contain less that 250kcal. I also query the values presented as breakfast energy consumption ( and the inter-quartile range?)
In discussing the strengths of the sample size I would encourage the inclusion of a statement regarding how many participants were required to reach statistical power.
Author Response
We would like to thank the editor and the reviewer for their useful comments which are all addressed in detail below. We hope that you will find revised manuscript suitable for publication in the Nutrients.
REPLY TO REVIEWERS’ COMMENTS
Though this is an interesting study, the limitation of 250kcal on what constitutes eating breakfast is potentially a methodological error. The methodology requires some expansion, though it is good to see that SES has been considered. The manuscript reaches a well considered conclusion and i am sure this paper will receive sufficient interest.
Authors: Thank you very much on your comments. We have addressed your concerns in details in our replies below.
Specific comments:
Abstract: CRO-PALS needs to be defined. Please specify that 't' is related to Sobel's 't' to provide clarity.
Authors: Thank you for your comment. We have defined CRO-PALS and specified Sobel's t clearly in the abstract and in the results where needed.
Please clarify how the cohort was recruited, the children approached to participate were determined by a cluster randomisation process? Please discuss if there were any inclusion or exclusion criteria associated with selection of the cohort, was there any consideration of dieting habits, disordered eating behaviours, health conditions, medications etc.
Authors: Thank you for your comment. Children approached to participate were determined by a cluster randomisation process and this has been further explained in the method section. Participants who had significantly reduced their food intake on the day prior to the dietary assessment due to health-related or any other reasons were excluded from the analysis.
Were any special considerations made to taking the anthropometric measurements of adolescents?
Authors: Thank you for your comment. No special considerations have been taken into account since everything was done according to suggested methodology. All participants were dressed appropriately as measurements were made during a physical education class. There were no cases when due to injury or some other reason we weren’t able to do any of the measurements as planned.
Were the skin fold measurements taken by a person with International Association of Kinanthropometry qualifications?
Abstract: Thank you for your comment. Skinfold measurements were taken by a single, skilled lab technician, anthropometrist with more than 30 years of experience in taking anthropometric measurements in Laboratory of Kinanthropometry, Dept. of sport Medicine, Faculty of Kinesiology.
Please add in evidence to the literature as to why the sum of four skin folds is considered an appropriate measure of adiposity.
Authors: Thank you for your comment. Of all anthropometric measures, skinfolds, and especially the sum of skinfolds, show the greatest correlation with body density. Therefore, the sum of skinfolds is usually used in the evaluation of adiposity, and the regression equations for assessing fat percentage include the sum of three or more skinfolds. This has been further clarified in the method section.
References added to the list of References:
Lohman, T.G.; Hingle, M.; Going, S.B. Body Composition in Children. Pediatric Exercise Science 2013, 25, 573–590.
Were parents consulted regarding the child's food recall to ensure accuracy?
Authors: The methods most commonly used to evaluate diet in adolescents are dietary records, the 24-h dietary recall and food frequency questionnaires. 24-h dietary recalls are most often used on large number of participants since they provide fast and accurate information of dietary intake. A study by Altbar et al. (doi:10.1017/S0007114516000593) has shown that only adolescents <12 years of age require assistance from the researcher or parents to obtain more accurate data, regardless of the dietary assessment method used. In older adolescents, trained interviewer using multi-pass 24-h recall is suitable to gain accurate data.
Why was an energy qualifier put on what constitutes breakfast? A healthy and filling breakfast can quite easily contain less that 250kcal. I also query the values presented as breakfast energy consumption (and the inter-quartile range?)
Authors: Among several different definitions of breakfast (O’Neil et al., 2014), we opted for a definition related to energy intake driven by suggestions for defining eating occasions using minimal energy intake (Leech et al, 2015). In Croatian dietary guidelines for chidren (Capak, 2015), a minimum of 10% of total energy intake is considered a meal instead of a snack. Since average energy requirements in adolescents are estimated at 2500 kcal/day, we set 250 kcal as the minimal energy intake to be taken on a single eating occasion.
Similarly, 250 kcal cut-off was used in previous studies, among others (https://doi.org/10.1093/advances/nmy047). This has been clarified in a method section as well.
References added to the list of References:
O’Neil, C.E.; Byrd-Bredbenner, C.; Hayes, D.; Jana, L.; Klinger, S.E.; Stephenson-Martin, S. The Role of Breakfast in Health: Definition and Criteria for a Quality Breakfast. Journal of the Academy of Nutrition and Dietetics 2014, 114, S8–S26.
Leech, R.M.; Worsley, A.; Timperio, A.; McNaughton, S.A. Characterizing eating patterns: a comparison of eating occasion definitions. Am J Clin Nutr 2015, 102, 1229–1237.
Capak, K.; Colić Barić, I.; Musić Milanović, S.; Petrović, G.; Pucarin- Cvetković, J.; Jureša, V.; Pavić Šimetin, I.; Pejnović Franelić, I.; Pollak, L.; Pavić, E.; et al. Nacionalne smjernice za prehranu učenika u osnovnim školama. 2015.
In discussing the strengths of the sample size I would encourage the inclusion of a statement regarding how many participants were required to reach statistical power.
Authors: Strengths of this study include a reasonably large sample size and inclusion of mediating variables such as PA, sleeping time, age and SES. Post-hoc analysis has been performed using statistical power analysis program G*Power for Windows 3 (Faul et al. 2007). It demonstrated this study was able to detect a medium effect size (f2=0.15; Cohen 1988, p.412) in both girls and boys with the power (1–β error probability) of >0.999, when setting the significance level at p=0.05. This statement has been added in discussion where appropriate.
References added to the list of References:
Faul, F.; Erdfelder, E.; Lang, A.-G.; Buchner, A. G*Power 3: A flexible statistical power analysis program for the social, behavioral, and biomedical sciences. Behavior Research Methods 2007, 39, 175–191.
Cohen, J. Statistical power analysis for the behavioral sciences; L. Erlbaum Associates: Hillsdale, N.J., 1988; ISBN 978-0-8058-0283-2.
Reviewer 2 Report
The authors habe obtained interesting result about the relations between breakfast consumption, adiposity measures and physical activity in a sample of adolescents. Whwn grouped by tertiles of physical activity, the association of breakfast consumption and lower adiposity was not present in low active groups of adolescents.Please consider this suggestion:
Abstracts
n=802,52% boys??
This investigation is based on 802 participants (48% girls and 52% boys).
The authors should consider remove the word "Encouraging" from the last sentence as breakfast is an essential meal for adolescents even if the do not perform adequate physical activity.
Results
It would have been interesting to compare boys with girls in this study. The authors indicate in Lines 219-222: "Only in the most active boys there was significant association between
breakfast consumption and lower adiposity (p=0.04). An identical pattern of moderation effect was seen also in girls, although slightly wider CI was noted, and no significant effect was noticed". Therefore only in boys has this effect been found and yet in the conclusions it extends to adolescents in general.
Discussion
I agree with the authors that future relevant studies that will use objective methods of physical activity assessment in large samples as well as additional confounders such as number of eating occasions and total daily energy intake are needed.
Author Response
Dear reviewer,
We would like to thank the editor and you for their useful comments which are all addressed in detail below. We hope that you will find revised manuscript suitable for publication in the Nutrients.
Reviewer comments:
The authors habe obtained interesting result about the relations between breakfast consumption, adiposity measures and physical activity in a sample of adolescents. Whwn grouped by tertiles of physical activity, the association of breakfast consumption and lower adiposity was not present in low active groups of adolescents.
Please consider this suggestion:
Abstracts
n=802,52% boys??
This investigation is based on 802 participants (48% girls and 52% boys).
Authors: Thank you for your comment. We have revised our abstract as suggested.
The authors should consider remove the word "Encouraging" from the last sentence as breakfast is an essential meal for adolescents even if the do not perform adequate physical activity.
Authors: Thank you for your comment. We agree with your statement and have removed the word “encouraging” from the last sentence accordingly.
Results
It would have been interesting to compare boys with girls in this study. The authors indicate in Lines 219-222: "Only in the most active boys there was significant association between
breakfast consumption and lower adiposity (p=0.04). An identical pattern of moderation effect was seen also in girls, although slightly wider CI was noted, and no significant effect was noticed". Therefore, only in boys has this effect been found and yet in the conclusions it extends to adolescents in general.
Authors: Thank you for your comment. We have changed conclusion taking into account your suggestion.
Discussion
I agree with the authors that future relevant studies that will use objective methods of physical activity assessment in large samples as well as additional confounders such as number of eating occasions and total daily energy intake are needed.
Round 2
Reviewer 1 Report
Thank you for addressing my comments.